# Structural basis for the regulation of plant transcription factor WRKY33 by the VQ protein SIB1
Xu Dong [1,2,6], Lulu Yu[3,4,6], Qiang Zhang[1,2,6], Ju Yang[1], Zhou Gong[1,2], Xiaogang Niu[4,5], Hongwei Li[4,5], Xu Zhang [1,2], Maili Liu [1,2], Changwen Jin [3,4,5] ✉ & Yunfei Hu [1,2] ✉

The WRKY transcription factors play essential roles in a variety of plant signaling pathways associated with biotic and abiotic stress response. The transcriptional activity of many WRKY members are regulated by a class of intrinsically disordered VQ proteins. While it is known that VQ proteins interact with the WRKY DNA-binding domains (DBDs), also termed as the WRKY domains, structural information regarding VQ-WRKY interaction is lacking and the regulation mechanism remains unknown. Herein we report a solution NMR study of the interaction between *Arabidopsis* WRKY33 and its regulatory VQ protein partner SIB1. We uncover a SIB1 minimal sequence neccessary for forming a stable complex with WRKY33 DBD, which comprises not only the consensus "FxxhVQxhTG" VQ motif but also its preceding region. We demonstrate that the $\beta_N$-strand and the extended $\beta_N$-$\beta_1$ loop of WRKY33 DBD form the SIB1 docking site, and build a structural model of the complex based on the NMR paramagnetic relaxation enhancement and mutagenesis data. Based on this model, we further identify a cluster of positively-charged residues in the N-terminal region of SIB1 to be essential for the formation of a SIB1-WRKY33-DNA ternary complex. These results provide a framework for the mechanism of SIB1-enhanced WRKY33 transcriptional activity.

WRKY proteins are one of the largest families of transcription factors (TFs) found almost exclusively in higher plants, with over 70 members identified in *Arabidopsis thaliana*. They play critical roles in plant resistance to various biotic and abiotic stresses, and are also implicated in the regulation of developmental processes[1–3]. The DNA-binding domains (DBDs) of WRKY TFs, also designated as the WRKY domains, contain a zinc-finger motif and a WRKYGQK consensus, and recognize a TTGACY (Y is C or T) W-box motif in gene promoter regions[4–6]. Based on the number of WRKY domains present and the zinc-finger pattern, WRKY proteins can be categorized into three major groups, namely groups I, II and III[1,2]. Evolutionary studies further suggested that group II is not monophyletic and splits up into five subgroups IIa-e[1,7]. Several structural studies showed that WRKY domains adopt a β-sheet structure, and the β-strand harboring the WRKYGQK consensus wedges into the DNA major groove during interaction[8–13]. The high conservation of WRKY domain sequences and the essentially identical DNA-binding interfaces raise the question of how the functional diversity and specificity of different WRKY members are regulated.

Studies of plant immune responses to pathogens like *Botrytis cinerea* or *Pseudomonas syringae* revealed that WRKY33, a group I WRKY TF member[1,7], acts downstream of the pathogen-responsive mitogen-activated protein kinases (MAPKs) MPK3/MPK6 and MPK4[14–16]. A growing body of evidence highlights a specific class of VQ proteins that function as transcriptional regulators of WRKY TFs[17–20]. For example, MKS1 (VQ21) forms a ternary complex with MPK4 and WRKY33 in the nucleus, and its phosphorylation by MPK4 when infected with *P. syringae* releases WRKY33 to bind to gene promoter regions[16]. Two homologous sigma factor-binding proteins SIB1 (VQ23) and SIB2 (VQ16) interact with the DNA binding domain of WRKY33. This leads to an increase in the DNA-binding ability of WRKY33 during host defense against *B. cinerea* infection[21]. The hallmark of group I WRKY TFs is the presence of two DBDs, the N-terminal DBD

[1]State Key Laboratory of Magnetic Resonance and Atomic and Molecular Physics, National Center for Magnetic Resonance in Wuhan, Innovation Academy for Precision Measurement Science and Technology, Chinese Academy of Sciences, Wuhan 430071, China. [2]University of Chinese Academy of Sciences, Beijing 100049, China. [3]College of Life Sciences, Peking University, Beijing 100871, China. [4]Beijing Nuclear Magnetic Resonance Center, Peking University, Beijing 100871, China. [5]College of Chemistry and Molecular Engineering and Beijing National Laboratory for Molecular Sciences, Peking University, Beijing 100871, China. [6]These authors contributed equally: Xu Dong, Lulu Yu, Qiang Zhang. ✉e-mail: changwen@pku.edu.cn; huyunfei@wipm.ac.cn

(nDBD) and the C-terminal DBD (cDBD). Only the cDBD is able to bind VQ proteins[17,18,21]. Up to date, about 34 VQ proteins have been identified in *Arabidopsis thaliana*. They all display sequence features characteristic of intrinsically disordered proteins (IDPs). Apart from the consensus "FxxhVQxhTG" VQ motif (where h is a hydrophobic residue and x is any residue), they show high sequence divergence[17,18,20]. They are classified into ten different groups and have distinct effects on WRKY transcriptional activity (e.g., activation, repression, or no effects)[17,19,20]. In order to unravel the complexities of plant stress-response signaling, understanding how VQ proteins selectively bind and regulate specific WRKY domains remains one of the key issues to be resolved. However, current knowledge of VQ-WRKY interactions were obtained mostly from in vivo genetic studies, whereas a structural-based understanding of the binding mechanism is completely lacking.

Herein, we employ the solution nuclear magnetic resonance (NMR) method to investigate the interaction between *Arabidopsis* SIB1 and WRKY33 cDBD (abbreviated as WRKY33_C hereafter). We identify the binding sites in both SIB1 and WRKY33_C and build a SIB1-WRKY33_C complex model based on NMR titration and paramagnetic relaxation enhancement (PRE) data. The model suggests a mechanism of how SIB1 may enhance WRKY33_C-DNA interaction, and helps uncover a lysine cluster in the N-terminal region of SIB1 to be essential for SIB1-WRKY33_C_DNA ternary complex formation. Our results offer a framework for VQ-WRKY interactions and provide insights into understanding how VQ proteins may regulate the diverse functions of WRKY TFs.

## Results
### Expression and functional characterization of SIB1
As our initial attempts to obtain soluble expression of full-length SIB1 failed, we screened a series of SIB1 constructs with a deletion in its N- and C-flanking regions without perturbing the VQ consensus. We successfully achieved large-scale soluble expression of the T11-L100 segment, designated as SIB1[11-100] (Fig. 1a). The deleted region in the N-terminal segment

constitutes a chloroplast targeting signal peptide. It has been demonstrated in previous studies that this segment is not essential for the functioning of SIB1 in plant defense against *B. cinerea* infection[21]. On the other hand, the C-terminal 51 residues show a noticeable increase in hydrophobicity (Supplemental Fig. S1), which could potentially explain the challenges encountered in achieving soluble expression.

To verify that the SIB1[11-100] construct retains its functional activity, we examined whether it is able to promote the DNA-binding activity of WRKY33_C, as previously reported for the full-length SIB121. Electrophoretic mobility shift assay (EMSA) experiments were carried out in the presence or absence of SIB1[11-100] at varying molar ratios between the W-box DNA and WRKY33_C. The results show that the presence of SIB1[11-100] strongly enhances the WRKY33_C-DNA binding, accompanied by the formation of a ternary complex (Fig. 1b). This suggests that the truncated construct retains the functional properties of the full-length protein. For brevity, we hereafter refer to the SIB1[11-100] construct as SIB1.

The in vitro stoichiometry binding between SIB1 and WRKY33_C was identified to be 1:1 by chemical cross-linking experiments (Fig. 1c and Supplemental Fig. S2a). This was further confirmed by size-exclusion chromatography of the complex formed between WRKY33_C and a Trx-SIB1 fusion protein (Fig. 1d and Supplemental Fig. S2b). The fusion with Trx does not perturb the interaction between SIB1 and WRKY33_C, as verified by NMR spectroscopy. The elution volume of the complex is in between the range of 31 to 43 kDa, supporting a 1:1 complex with a molecular weight of ~35 kDa.

### Conformational changes of SIB1 upon binding to WRKY33_C
To understand how the intrinsically disordered SIB1 binds WRKY33_C, we prepared [15]N-labeled SIB1 in both its free and WRKY33_C-complexed states. The two-dimensional (2D) [1]H-[15]N heteronuclear single quantum coherence (HSQC) NMR experiments were used to monitor its binding and structural changes (Fig. 2a). The spectrum of free SIB1 shows the clustering of sharp signals in the central region with very narrow chemical shift

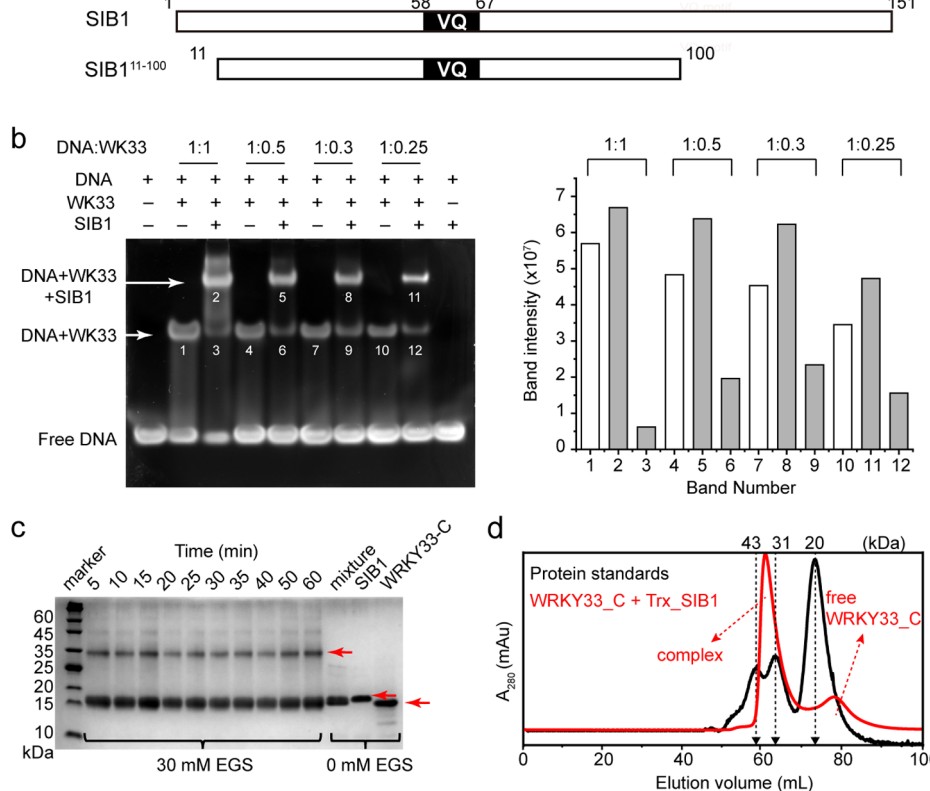

**Fig. 1 | In vitro interaction between SIB1 and WRKY33_C. a** An illustration of the soluble SIB1[11-100] truncation construct. **b** Gel electrophoresis of the binding between W-box DNA and WRKY33_C under different molar ratios in the presence or absence of SIB1 (left) and the band intensity read-outs corresponding to binary or ternary complexes (right). WRKY33_C is designated as WK33 for short. **c** SDS-PAGE analysis of the chemical cross-linking results between WRKY33_C and SIB1. The bands corresponding to the free WRKY33_C, SIB1, and the cross-linked complex are indicated by red arrows (note that both SIB1 and WRKY33_C migrate at apparent molecular weights much larger than their theoretical values). **d** Size-exclusion chromatography profile showing the complex formation between WRKY33_C and Trx-SIB1. The profile of the mixed sample of Trx_SIB1 and WRKY33_C is shown in red, and that of protein standards for molecular weight calibration is shown in black.

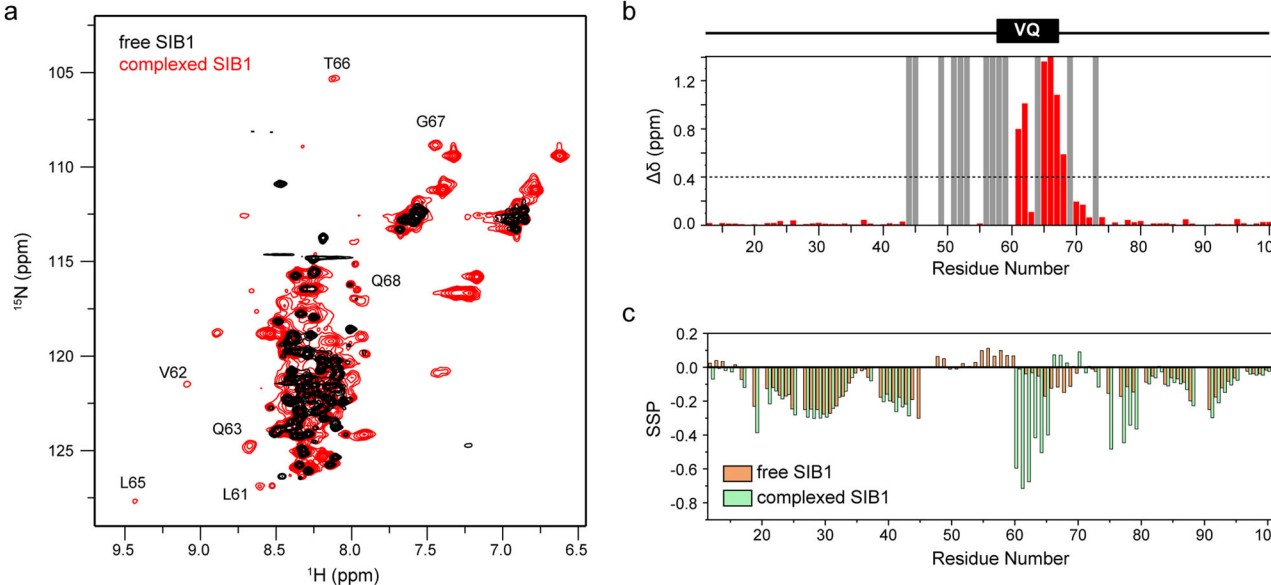

**Fig. 2 | NMR analyses of SIB1 bound to WRKY33_C. a** Overlay of the 2D $^{1}$H-$^{15}$N HSQC spectra of $^{15}$N-labeled SIB1 in the free form (black) or in complex with unlabeled WRKY33_C (red). A few representative well-dispersed signals newly appeared in the complexed state are annotated with their assignments. **b** Chemical shift differences between free and complexed states of SIB1. The composite chemical shift changes ($\Delta\delta$) were calculated using the empirical equation

$\Delta\delta = \sqrt{\Delta\delta_H^2 + (\Delta\delta_N/6)^2}$, where $\Delta\delta_H$ and $\Delta\delta_N$ are the chemical shift changes in the $^{1}$H and $^{15}$N dimensions, respectively. Residues unassigned only in the complexed state are indicated by gray bars. **c** The SSP scores of SIB1 in the free and complexed states calculated based on all available $C^{\alpha}$, $C^{\beta}$, and $H^{N}$ chemical shifts.

dispersion (~1 ppm in the $^{1}$H dimension). This reflects highly similar chemical environments for the polypeptide backbone amide groups, indicating that the free SIB1 adopts an overall unfolded conformation. In the WRKY33_C-complexed state, although most signals remain clustered, several well-dispersed new peaks with $^{1}$H chemical shifts in the 8.5–9.5 ppm region are observed. This suggests that binding to WRKY33_C induces local secondary structure formation in SIB1.

To identify which segment of SIB1 undergoes binding-induced folding, we collected the conventional triple-resonance NMR experiments using $^{13}$C/$^{15}$N-labeled SIB1 in both the free and complexed states to acquire the backbone chemical shift assignments. For the free SIB1, the backbone amide resonances could be assigned for 72 out of 79 non-proline residues, whereas for the complexed form, 60 out of 79 non-proline residues were assigned (Supplemental Fig. S3). Backbone chemical shift perturbation (CSP) analysis shows that the 60–70 segment harboring the VQ motif is the most heavily affected (Fig. 2b). In particular, the newly appeared well-dispersed peaks indicating folded structures mostly originate from residues Q63-G67. Furthermore, by using the secondary structural propensity (SSP) method based on combined analysis of the secondary chemical shifts of backbone atoms[22], we found that the L61-T66 segment gains obviously increased propensities of forming extended β-strand-like conformation (Fig. 2c). These observations highlight the central role of the 60–70 segment of SIB1 in interacting with WRKY33_C. Additionally, it is suggested that this particular segment of SIB1 may adopt a locally folded conformation upon binding. This is consistent with the previously reported observation that a VQ-deleted mutant of SIB1 failed to interact with WRKY33 by coimmunoprecipitation analysis[21].

Apart from the VQ motif, two additional segments show distinct changes upon binding to WRKY33_C. One is the 40–60 segment preceding the VQ motif, a highly potential contributor to the binding. The majority of residues in this region disappear or become too weak to be confidently assigned in the complexed state. This implies conformational exchanges occurring on an intermediate NMR timescale, which results in peak broadening. (Fig. 2b). The other is the 76–80 segment, which shows very slight backbone amide chemical shift changes, but displays changes in the SSP scores, indicative of a β-forming trend in the complexed state (Fig. 2c).

### The minimal sequence of SIB1 required for WRKY33_C binding

During our initial attempt to obtain three-dimensional structural information of the SIB1-WRKY33_C complex, we tried using a decapeptide comprising the essential VQ motif (SIB1$^{58-67}$: FRELVQELTG) to bind WRKY33_C. However, NMR titration of the peptide into a $^{15}$N-labeled WRKY33_C sample fails to induce obvious spectral changes similar to those observed with SIB1$^{11-100}$, albeit some signal disappearance occurs (Supplemental Fig. S4a). This suggests that the VQ motif alone is insufficient to bind WRKY33_C.

We therefore synthesized a series of peptides corresponding to different lengths and regions of SIB1, and their abilities to bind WRKY33_C were examined using 2D NMR (Fig. 3a, b and Supplemental Fig. S4b–d). Results show that neither the longer VQ-containing pentadecapeptide SIB1$^{55-69}$ nor the peptide composing its N-terminal neighboring region SIB1$^{40-60}$ can cause obvious changes in the 2D HSQC spectra of $^{15}$N-WRKY33_C, even with a fivefold excess of the peptides. In contrast, by using peptides spanning both regions (e.g., SIB1$36-69$ or SIB1$^{46-69}$), obvious spectral changes similar to SIB1$^{11-100}$ are observed (Fig. 3b and Supplemental Fig. S4d). These results demonstrate that the simultaneous binding of both segments is necessary for the formation of the SIB1-WRKY33_C complex.

Among the peptides capable of inducing $^{15}$N-WRKY33_C spectral changes, SIB1$^{46-69}$ contains the minimal number of residues, so it is defined as the minimal SIB1 construct (SIB1$^{mini}$) for interaction with WRKY33_C. This construct comprises two parts: the highly conserved VQ motif, which shows the largest chemical shift changes upon binding, and its preceding segment, which becomes mostly unobservable upon binding. The CSP profile of $^{15}$N-WRKY33_C induced by binding to SIB1$^{mini}$ closely resembles that induced by SIB1$^{11-100}$ (Fig. 3c and vide infra). This indicates that the core interacting site resides in the 46–69 segment of SIB1.

### The β$_N$ strand and β$_N$-β$_1$ loop of WRKY33_C form the SIB1-docking site

To identify the SIB1-binding site on WRKY33_C, we completed the backbone chemical shift assignments of WRKY33_C in both its free and SIB1-bound states (Supplemental Fig. S5). Upon binding to SIB1 or SIB1$^{mini}$, severe signal disappearance is observed for $^{15}$N-WRKY33_C, particularly in

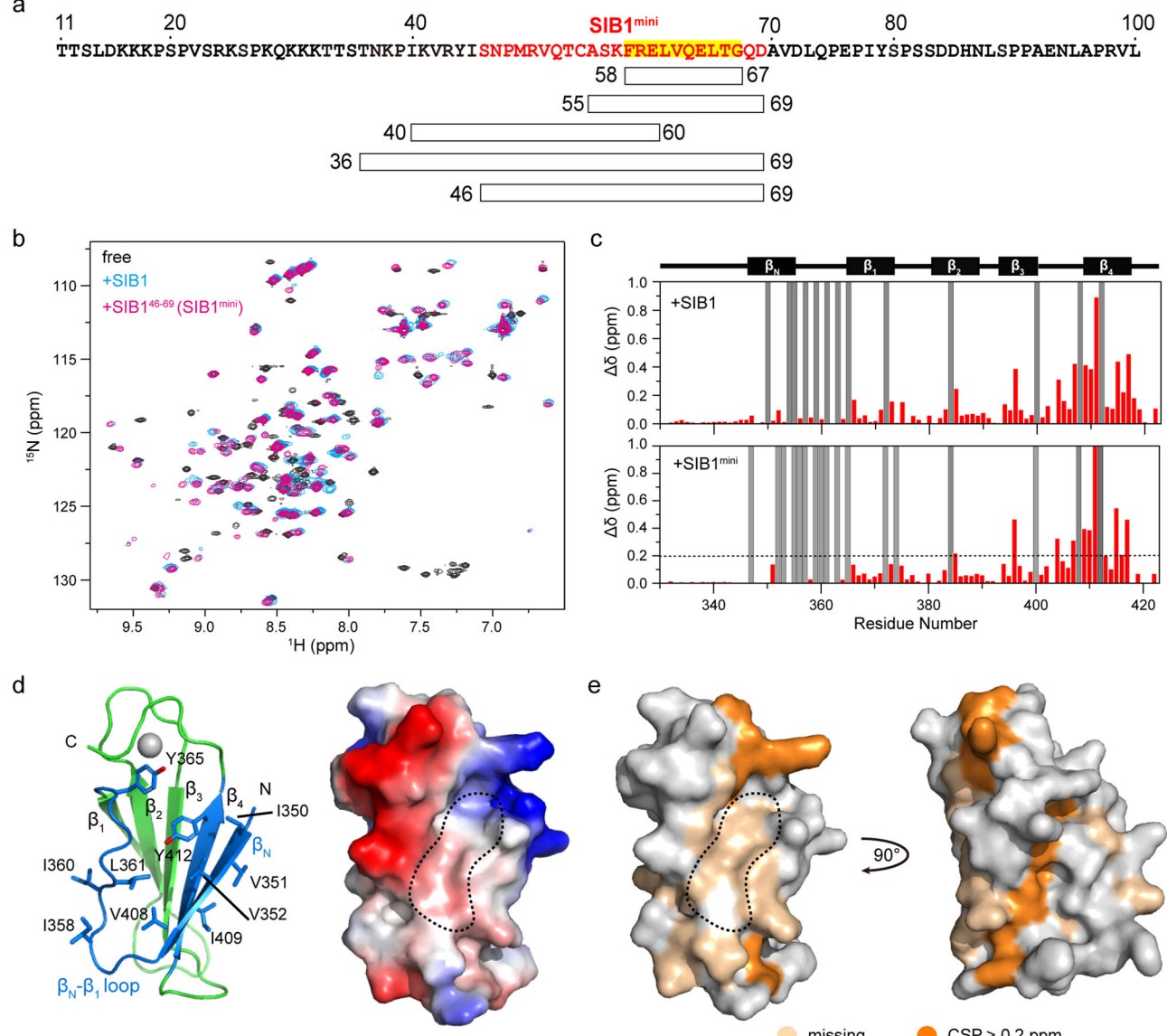

**Fig. 3 | Identification of SIB1-WRKY33_C binding sites. a** Schematic illustration of a series of SIB1-derived peptides used to identify the minimal sequecne required for WRKY33_C interaction. The sequence corresponding to SIB1$^{mini}$ is colored in red, and the consensus VQ motif is highlighted. **b** The $^1$H-$^{15}$N HSQC spectra of WRKY33_C in the free state (black) and in complex with SIB1 (blue) and SIB1$^{mini}$ (magenta). **c** The CSP profiles of the $^{15}$N-labeled WRKY33_C upon titration of SIB1 or SIB1$^{mini}$. Gray bars indicate residues that are missing the complexed state. **d** Structural model of WRKY33_C shown as the cartoon diagram (left) and the surface representation showing the charge distribution (right). **e** Mapping of the obviously perturbed residues onto the WRKY33_C surface representation.

the $\beta_N$ strand and the long $\beta_N$-$\beta_1$ loop (Fig. 3c). In addition, residues in and close to the $\beta_4$ strand, which packs adjacent to the $\beta_N$, show the largest CSP values. By mapping both the missing and perturbed residues onto the WRKY33_C structure, we found that they form a continuous surface located at one edge of the β-sheet (Fig. 3e). Notably, many of the missing residues located in a shallow solvent-exposed pocket formed between the $\beta_N$-$\beta_1$ loop and the $\beta_N$ stand. This region is rich in residues with hydrophobic or aromatic sidechains, such as V408-I409, Y412, I350-V352, I358, I360-L361, and Y365 (Fig. 3d). The local hydrophobicity likely facilitates interaction with the hydrophobic residues of SIB1$^{mini}$.

Among the available WRKY domain structures, not all contain five β-strands. For example, β-sheet structures containing only four strands were reported for the N-terminal DBD of Arabidopsis WRKY1, WRKY2, and WRKY33, all of which belong to the group I WRKY family[13]. Although the absence of the $\beta_N$ strand neither destabilizes the protein structure nor disrupts DNA binding, our results suggest that it is important for forming the

SIB1-binding site. We prepared a $\Delta\beta_N$ truncated construct of WRKY33_C comprising only the V352-A422 region. The 2D $^1$H-$^{15}$N NMR spectrum of this mutant exhibits well-dispersed signals, signifying a well-folded structure. However, titration with SIB1 fails to induce noticeable spectral changes. (Supplemental Fig. S6). This observation supports the key role of the $\beta_N$ strand in binding SIB1. Also, it is consistent with the previous observation that the N-terminal deletion of WRKY33_C disrupts its binding with SIB1[21].

**Determining SIB1-WRKY33_C binding mode by intermolecular PRE**

The severe loss of signals in both SIB1 and WRKY33_C upon binding implies that the interaction is highly dynamic, rendering it impossible to obtain conventional NOE restraints for the determination of an accurate complex structure. We, therefore, employed the PRE method, which relies on the relaxation enhancement effect on nuclear spins induced by the presence of a spin label (the paramagnetic center), to obtain long-range

**Fig. 4 | Inter-molecular PRE profiles of SIB1 induced by spin-labeled WRKY33_C.** The $\Gamma_2$ values of SIB1 induced by spin-labeled WRKY33_C at the D357C (**a**), R366C (**b**), and K376C (**c**) sites. The corresponding paramagnetic WRKY33_C samples are shown as cartoons on the right side of the $\Gamma_2$ plots, respectively. The paramagnetic tags on D357C, R366C, and K376C are shown as sticks, and the spin center ($Mn^{2+}$) are shown as sphere.

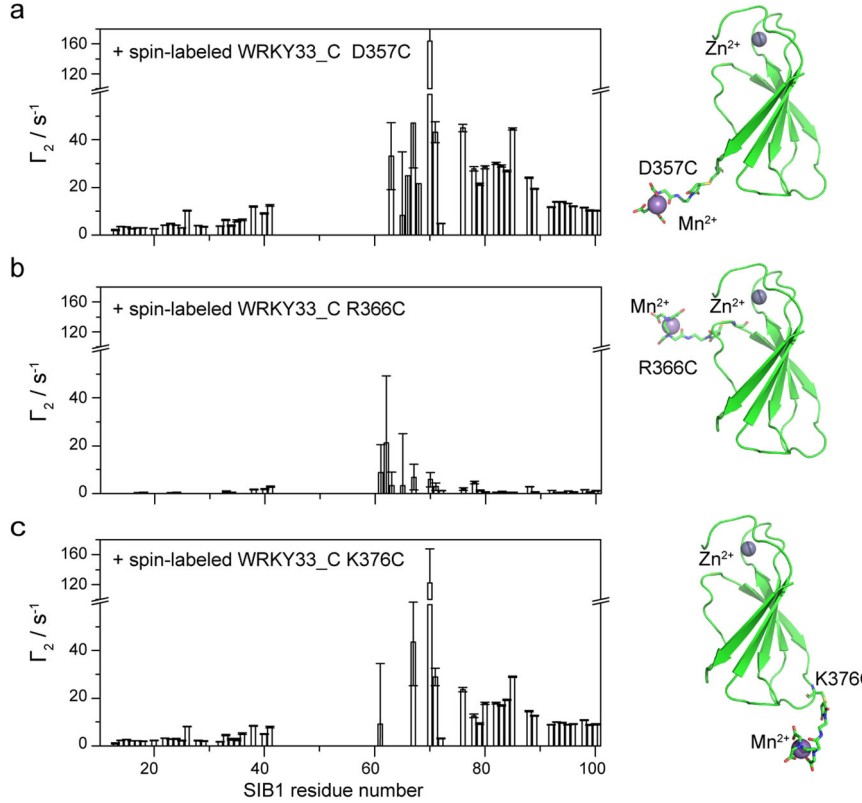

intermolecular distance restraints[23]. Three different sites, D357, R336, and K376, were individually mutated to cysteine for spin labeling. These sites are located at the C-terminal end of the $\beta_N$ strand and the N- and C-terminal ends of the WRKYGQK motif-containing $\beta_1$ strand, respectively (Fig. 4). The samples of $^{15}$N-labeled SIB1 in complex with WRKY33_C mutants or with $Ca^{2+}$-chelated WRKY33_C mutants show essentially similar HSQC spectra compared with that in complex with native WRKY33_C (Supplemental Figure S7-8). This verifies that spin labeling at these sites do not disturb the interaction between SIB1 and WRKY33_C.

As shown in Fig. 4, spin labeling at both D357C and K376C sites produces obviously higher transverse PREs ($\Gamma_2$) in the carboxyl region of SIB1, spanning the whole 60–100 segment. The PRE effects induced by labeling at the D357C site are stronger than K376C, and the most affected residues are located in the 60–85 segment of SIB1. This observation is consistent with the results from our CSP and SSP analyses. In contrast, attaching the paramagnetic center at the R366C site only results in small PRE effects only in the 60–70 segment of SIB1. In all three cases, the 40–60 segment is missing and cannot be analyzed, and the N-terminal part of SIB1 is minimally affected. These observations support a scenario that the C-terminal region of SIB1 binds onto the WRKY33_C protein surface, while the N-terminal region remains mobile.

Based on the observed PRE profiles, we are able to deduce the following characteristics for the SIB1-WRKY33_C binding pattern: (1) The 60–70 segment of SIB1, which corresponds to the VQ motif, forms the central interacting site, as it is perturbed by spin labeling in all three sites. (2) Residue 70, which shows the largest $\Gamma_2$ values in both the D357C and K376C spin-labeled samples, is expected to be located in the lower region (or the zinc-finger distal region) of the WRKY33_C structure (as shown in Fig. 4) in proximity to both sites. (3) Because the 60–70 segment of SIB1 displays gradually increasing $\Gamma_2$ values in both the D357C and K376C spin-labeled samples, while it shows small but apparently decreasing $\Gamma_2$ values in the R366C spin-labeled sample, this motif is expected to be oriented with its N-terminus in the zinc-finger proximal region and its C-terminus in the zinc-finger distal region of the WRKY33_C structure.

### Building the structure model of the SIB1-WRKY33_C complex

To obtain a more intuitive understanding of the binding, we built a model of the SIB1-WRKY33_C complex by performing molecular dynamics (MD) simulation guided by the PRE-derived distance restraints. To facilitate calculation, only the S46–S80 segment of SIB1 was used. This segment comprises the essential binding sequence S46–A70, and it also incorporates the subsequent ten residues that exhibit both SSP score changes and PRE effects in the D357C and K376C spin-labeled samples. (Figs. 2c, 4).

During our initial attempts of model building, PRE-derived inter-protein distance restraints were only added for the 61–70 segment of SIB1, whereas no distance constraints were available for the 46–59 segment due to their signal disappearance in the complex. Several rounds of independent MD simulations lasting 500 ns were run, and the resultant structural ensembles show a converged packing of the SIB1 L61-T66 segment onto the side of the WRKY33_C $\beta_N$ strand. Considering the presence of several hydrophobic residues in the L61-T66 segment, we propose that their contact with the protruding hydrophobic surface of the WRKY33_C $\beta_N$ strand may stabilize the interaction. Indeed, single site mutation of hydrophobic residues in the $\beta_N$ strand (I350A or V351A) leads to incomplete complex formation, as evidenced by the presence of a substantial fraction of free state resonances in the presence of excess SIB1$^{mini}$ (Supplemental Fig. S9).

The N-terminal region of SIB1, despite its indispensability for complex formation, is unconstrained and thus adopts varying conformations. However, when we closely examine the subset of conformers in which the SIB1 40–60 segment contacts the WRKY33_C surface, we observe that the SIB1 V51 is nearly always in proximity with two isoleucine residues (I358 and I360) in the WRKY33_C $\beta_N$-$\beta_1$ loop. Intrigued by this finding, as well as the previously reported observation that the V51 site is always occupied by a hydrophobic residue in other VQ proteins[18], we were curious to know whether such interactions truly exist.

To obtain experimental evidences, we prepared a SIB1$^{mini}$-V51A mutant peptide as well as a WRKY33_C-I358A/I360A double mutant protein. Neither the titration of SIB1$^{mini}$-V51A into $^{15}$N-labeled WRKY33_C nor the titration of SIB1$^{mini}$ into $^{15}$N-labeled WRKY33_C-I358A/I360A

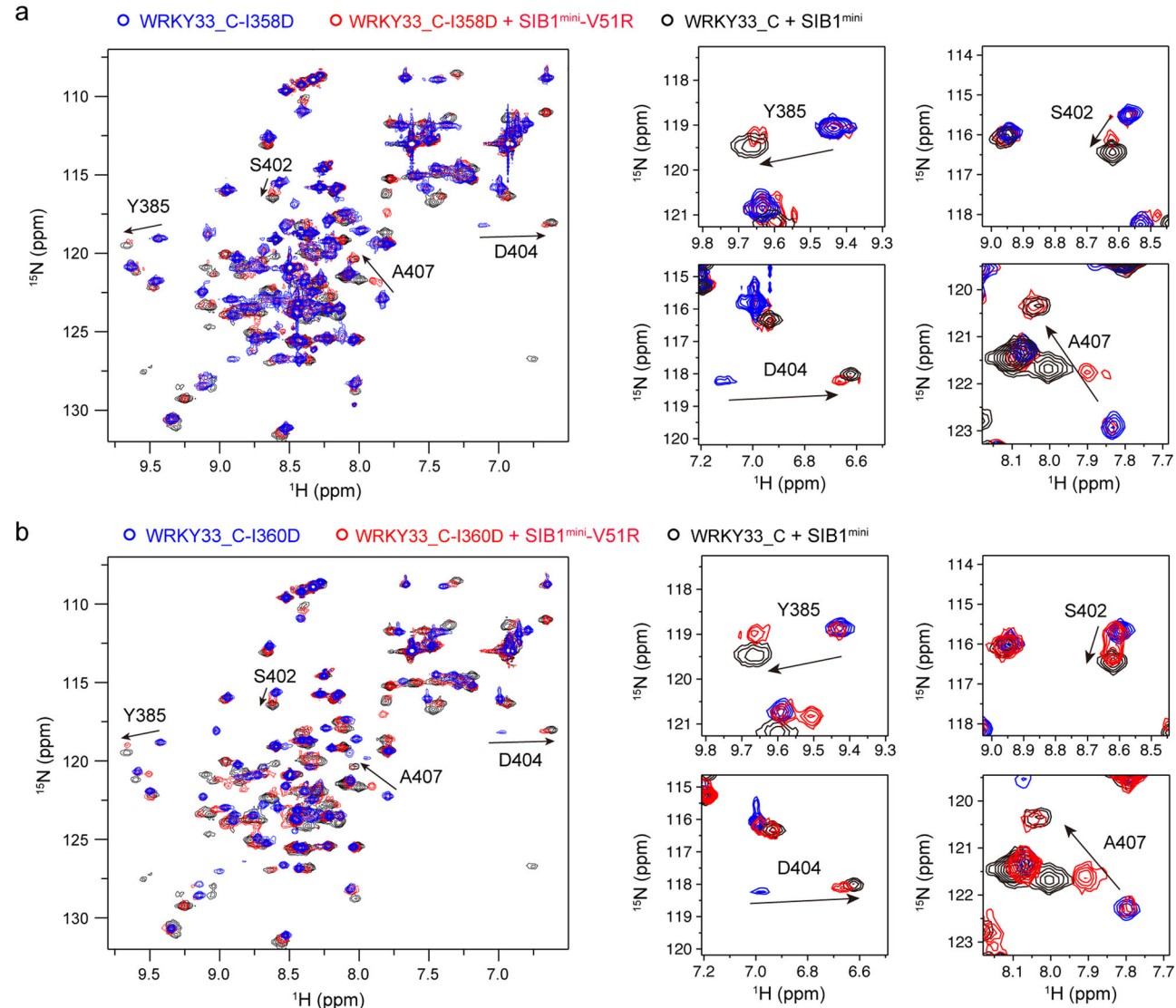

**Fig. 5 | Mutations to complementary charged residues at the I358/I360 site in WRKY33_C and V51 in SIB1 restore interaction. a, b** Overlay of the $^1$H-$^{15}$N HSQC spectra of $^{15}$N-labeled WRKY33_C-I358D (**a**) or WRKY33_C-I360D (**b**) mutants in their free states (blue) and in the presence of the SIB1$^{mini}$-V51R mutant peptide (red).

The spectrum of $^{15}$N-labeled wild-type WRKY33_C in complex with wild-type SIB1$^{mini}$ peptide is shown for comparison (black). Enlarged view of the spectral changes for representative resonances are shown on the right.

could induce obvious $^1$H-$^{15}$N HSQC spectral changes indicative of complex formation (Supplemental Fig. S10). This demonstrates the essential roles of these hydrophobic residues in stabilizing the SIB1-WRKY33_C complex. Additionally, we generated a SIB1$^{mini}$-V51R mutant peptide and two WRKY33_C mutant proteins in which either one of the two isoleucines were mutated to an aspartate (WRKY33_C-I358D and WRKY33_C-I360D). We hypothesized that if the above speculation is correct, the electrostatic interactions between the arginine and aspartate residues could, at least partially, facilitate the restoration of complex formation. As expected, the $^1$H-$^{15}$N HSQC spectra of $^{15}$N-labeled WRKY33_C-I358D (or WRKY33_C-I360D) show changes upon titration of the SIB1$^{mini}$-V51R peptide that are essentially similar to the wild-type samples (Fig. 5). These results provide us an additional inter-protein distance restraint involving the N-terminal region of SIB1$^{mini}$ that can be included in the MD simulations.

Consequently, both PRE- and mutagenesis-derived distance restraints were used in the final MD simulations. All structural snapshots in the simulation trajectories were analyzed, and 20 conformers showing the best correlation with the experimentally observed data were selected to represent the working model of the SIB1-WRKY33_C complex (Fig. 6 and Table 1).

The back-calculated theoretical PRE values based on these conformers show correlation coefficient $R = 0.84 \pm 0.01$ with the experimental data (Supplemental Fig. S11).

In this model, SIB1$^{mini}$ packs onto WRKY33_C surface in a hook-like conformation. The V51-K57 segment of SIB1 fits into the shallow groove formed between the $\beta_N$ strand and the $\beta_N$-$\beta_1$ loop of WRKY33_C, whereas the L61-T66 segment forms an extended structure and is packed onto the $\beta_N$ strand. The $F_{58}R_{59}E_{60}$ tripeptide forms a turn like structure linking the above two segments. We observe that in all the conformers, the sidechain of F58 packs against the WRKY33_C binding groove. The positively charged sidechains of residues K57 and R59 are always oriented towards the patch of negative charges on the WRKY33_C surface, while the negatively charged residues, such as E60, E63, D69, and D72, are placed in proximity with positively charged areas of WRKY33_C (Fig. 6b). These interacting features are generally conserved among the ensemble of conformers, suggesting that charge complementarity may help to correctly orient the intrinsically disordered SIB1 polypeptide and to facilitate complex formation. In addition, we observe that the V51-S56 segment forms a short helical structure in many of the conformers. This is consistent with the SSP analysis results of SIB1,

**Fig. 6 | Structure model of the SIB1-WRKY33_C complex. a** The ensemble of 20 representative conformations of SIB1[46–80] in complex with WRKY33_C. The S46-K57, F58-G67, and Q68-S80 segments of SIB1 are colored in cyan, red, and wheat, respectively. The sidechains of the SIB1 V51 residue and the WRKY33_C I358/I360 residues are shown in sticks. **b** A lowest-energy conformer of the SIB1-WRKY33_C complex, with the SIB1 peptide shown as cartoon and the WRKY33_C protein shown as a surface representation colored with electrostatic distribution.

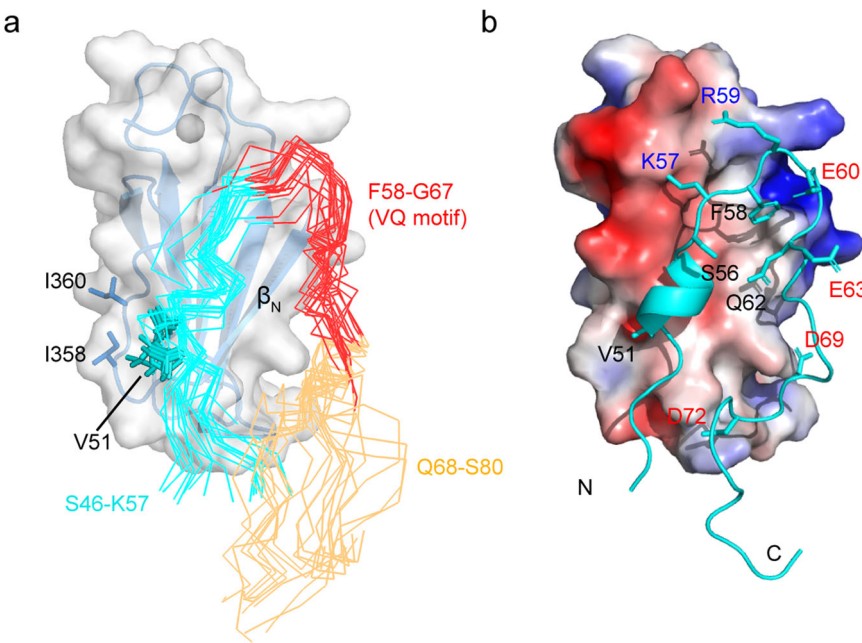

## Table 1 | NMR and refinement statistics for protein structures

|  | SIB1-WRKY33_C |
| --- | --- |
| **NMR distance and dihedral constraints** |  |
| Distance constraints | 19 |
| Total PRE | 17 |
| D357C spin-labeled | 8 |
| K376C spin-labeled | 9 |
| Mutagenesis-derived distance restraints[a] | 2 |
| Short-range (<10 Å) | 0 |
| Medium-range (10–20 Å) | 4 |
| Long-range (>20 Å) | 15 |
| Hydrogen bonds | 0 |
| Total dihedral angle restraints | 0 |
| $\phi$ | - |
| $\psi$ | - |
| **Structure statistics** |  |
| Correlation (mean and s.d.) |  |
| Number of conformers | 20 |
| Between experimental data and calculated data | 0.85 ± 0.01 |
| Average pairwise r.m.s. deviation** (Å) |  |
| Cα only | 1.83 ± 0.81 |
| Backbone | 1.86 ± 0.80 |
| Ramachandran statistics (only for SIB1 subunit) |  |
| Residues in most favored regions | 77.8% |
| Residues in additional allowed regions | 20.7% |
| Residues in generously allowed regions | 0.3% |
| Residues in disallowed regions | 1.2% |

**Pairwise r.m.s. deviation was calculated among 20 refined structures. Only the structures of the SIB1 subunit were involved in the calculation of pairwise r.m.s. deviation.

[a] The mutagenesis-derived distance restraints between WRKY33 I358/I360 and SIB1 V51 are set as 12.5 ± 0.5 Å

which suggests a slight helix-forming tendency in its free state, although data of the complexed state is not available due to signal disappearance (Fig. 2c).

### A lysine cluster in the SIB1 N-terminus is essential for enhancing WRKY33_C-DNA binding

The structure model of the SIB1-WRKY33_C complex provides a molecular basis for understanding how SIB1 may modulate the DNA-binding activity of WRKY33_C. Firstly, the SIB1- and DNA-binding interfaces are located on distinct sides of WRKY33_C without overlapping areas, and therefore the two binding events are not competitive with each other. Secondly, the binding orientation of SIB1 allows its flexible N-terminal region to be placed in a space proximal to the DNA-binding WRKYGQK motif of WRKY33_C. Because there are clusters of positively charged residues in the SIB1 N-terminal region, we speculate that it may form additional contacts with the negatively charged DNA to enhance the binding.

To test this hypothesis, we generated two SIB1 mutants in which two clusters of lysines in the N-terminal region were mutated to alanines separately (Fig. 7a). EMSA experiments show that while the cluster I K-to-A mutant (I-K2A) is able to promote the formation of a stable SIB1-WRKY33_C-DNA ternary complex similar to the wild-type SIB1, the cluster II K-to-A mutant (II-K2A) mutant fails (Fig. 7b). The highly smered band observed for II-K2A suggests that the ternary complex is unstable. Hence, the lysine residues in cluster II do not participate in the SIB1-WRKY33_C interaction; instead, they make vital contributions to engaging with the DNA molecule and stabilizing the SIB1-WRKY33_C-DNA ternary complex.

## Discussion

In this study, we elucidated the structural basis of the interaction between the plant VQ protein SIB1 and the WRKY33 transcription factor. Firstly, we identified the minimal sequence of SIB1 required for binding WRKY33_C and highlighted the indispensable role of the additional segment preceding the consensus VQ motif. Secondly, we identified the SIB1-binding site in the WRKY domain to be formed by the $\beta_N$ strand and $\beta_N$-β1 loop, which are neither required for DNA binding nor present in all WRKY domains. This provides an explanation for why VQ proteins bind to only a subset of WRKY domains. Thirdly, to overcome intrinsic dynamics, we integrated multiple experimental methods with MD simulations to generate a structural model of the SIB1-WRKY33_C complex. This model, for the first time,

**Fig. 7 | Identification of a lysine cluster in the SIB1 N-terminus essential for enhancing WRKY33_C-DNA binding. a** Location of two clusters of lysine residues in the N-terminal region of SIB1. **b** EMSA experiments monitoring the SIB1-enhanced WRKY33_C-DNA interaction.

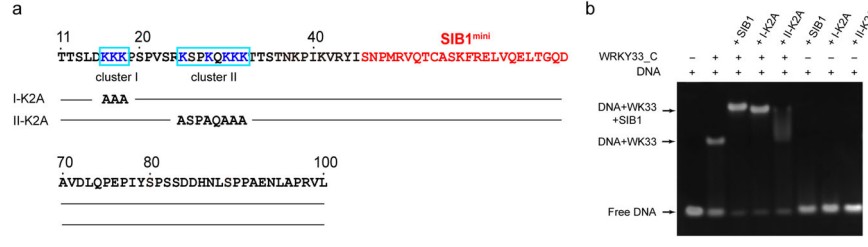

offers a molecular basis for understanding how VQ proteins bind to and regulate WRKY domains.

Currently, at least five VQ members in Arabidopsis have been proposed to interact with WRKY33, and they exhibit distinct effects on DNA binding. These members include SIB1 (VQ23), SIB2 (VQ16), MKS1 (VQ21), VQ4, and VQ10[17,18]. While SIB2 is homologous to SIB1, the other three show much less sequence similarity (Supplemental Fig. S12). Notably, the segment showing the highest similarity among the five proteins corresponds to the sequence V51-G67 in SIB1, which is exactly the central part of the SIB1[mini] sequence identified to form core interactions with WRKY33_C upon complex formation. In particular, residue V51 is highly conserved among the five proteins, implicating that the interaction between this valine and the isoleucine residues on the WRKY33_C surface may be conserved in other WRKY33-VQ complexes. In contrast, the five proteins show large sequence divergence in both the N- and C-terminal flanking regions of this consensus sequence. Unlike SIB1 and SIB2, both of which display transcription activation activity in the previously reported transient expression assay, VQ4 and MKS1 both repress transcription activity, whereas VQ10 shows no obvious effect[17,24]. It is possible that a lack of or a distinct location of clusters of positively charged residues in the N-flanking region may contribute to their differential regulatory functions.

Taken together, our study provides a structural model of the interaction between WRKY domains and the VQ proteins. In this model, the intrinsically disordered VQ proteins anchor onto WRKY domain surfaces via its central segment that comprises both the consensus VQ motif and its preceeding sequence, whereas the diverse N- and C-regions can fulfill distinct regulatory functions in gene-specific transcriptions[17,18]. Consistent with our results, experimental evidence supporting the importance of the segment preceding the VQ motif has also been reported for soybean VQ proteins[25].

In addition, AlphaFold2[26] was also used to predict the structure of the SIB1-WRKY33_C complex. Although the results predicted the same interacting surface on the WRKY33 C-domain, details in the packing orientations or local secondary structure formations are completely different from the experimentally derived model (Supplemental Fig. S13). In particular, the AlphaFold2-predicted model designates the F58-G67 segment to adopt a helical conformation, and it also assigns the preceding S46-K57 segment to form a β-strand, packing to the side of the WRKY33_C β_N strand. These predictions are not in agreement with either the PRE data, the chemical shift-based secondary structural analysis, or the mutagenesis results associated with residue V51. Nevertheless, we cannot entirely exclude the possibility that the AlphaFold2-predicted model may represent one possible way of SIB1-WRKY33_C complex formation, since the NMR data suggest strong dynamics. However, such a conformational state is likely to be too sparse (if it exists) to be captured by the currently used methods.

In the SIB1-WRKY33_C complex, the lysine cluster in the flexible SIB1 N-terminus may aid in the formation of a relatively stable SIB1-WRKY33_C-DNA ternary complex, thereby regulating transcriptional activity. To elucidate how this lysine cluster may facilitate ternary complex formation, we extended our analysis by constructing a model of the SIB1-WRKY33_C-DNA ternary complex. This model was based on the structure of the SIB1-WRKY33_C binary complex and the crystal structure of the complex formed between the N-terminal DNA binding domain (DBD) of AtWRKY1 and its DNA partner (PDB entry: 6J4E)[13]. To account for the

contribution of the cluster II lysine residues in SIB1 to DNA binding, a radius of gyration restraint was added for these residues together with the whole DNA molecule. Among the calculated structures, we observe two major classes of conformations in which the flexible SIB1 N-terminus extends over to wrap around the DNA duplex from two different sides (Supplemental Fig. S14). The model indicates that the sequence length between the cluster II lysines and the SIB1[mini] segment closely matches the physical size of the WRKY33_C-DNA complex. This allows SIB1 to string the two binding partners together and may be favorable for the stabilization of the ternary complex.

Based on the EMSA results, we could estimate that the binding between WRKY33_C and the W-box DNA has a dissociation constant $K_d$ in the sub-nanomolar range (~150 nM). However, in the case of the binding between the SIB1-WRKY33_C binary complex and DNA, the binding curves deviate from a two-state exchange model probably due to the partial dissociation of the SIB1-WRKY33_C complex (Fig. 1b and Supplemental Fig. S15). Furthermore, we noted that the SIB1-WRKY33_C-DNA ternary complex tends to precipitate at higher concentrations (e.g., sub-millimolar to millimolar range), thereby preventing the use of isothermal titration calorimetry or solution NMR techniques for further characterization of the binding. Therefore, we were not able to provide an accurate estimation of the binding affinity under the current experimental conditions. While our current study offers a structural-based hypothesis for the function of the cluster II lysines in promoting a ternary complex formation, we anticipate that future in vivo studies, e.g. in vivo transcriptional activity assays, or functional examinations of Arabidopsis resistance to B. cinerea infection using transgenic plants harboring SIB1/2 mutants as described by ref. 21, to be essential in providing further insights into the physiological role of the lysine cluster.

Recently, SIB1 and SIB2 have also been found to interact with Arabidopsis WRKY75, a group IIc member, and act in the abscisic acid (ABA)-mediated leaf senescence leaf senescence and seed germination pathways[27]. The reported results suggest that SIB1/2 downregulates the transcriptional repression activity of WRKY75. By comparing the structural model of the WRKY75 DNA-binding domain with the SIB1-WRKY33_C complex structure, we observe that WRKY75 exhibits a generally similar electrostatic distribution pattern as WRKY33_C. Additionally, WRKY75 contains two hydrophobic residues, Val63 and Ile65, at positions corresponding to I358 and I360 of WRKY33 (Supplemental Fig. S16). These similar characteristics suggest that SIB1/2 may also interact with WRKY75 at the same site and probably via an analogous binding pattern. However, the exact binding mode, as well as the mechanisms of how SIB1/2 contributes to the regulation of different signaling pathways, remain to be investigated. We anticipate that further structural and functional studies can provide a more comprehensive understanding of how different VQ-WRKY pairs interact with each other, and how the diverse N- and C-flanking sequences function in the complex transcriptional regulation network in plant stress responses.

## Methods
### Protein expression and purification
Genes encoding Arabidopsis SIB1 protein and WRKY33_C domains were cloned into the pET-21a(+) vector (Novagen) with a C-terminal 6×His-tag. All constructs were transformed into the Escherichia coli BL21(DE3) strain (Sigma-Aldrich) for protein expression. The cell cultures were first grown in 1 L of Luria–Bertani (LB) broth medium at 35 °C with 100 mg/mL of

ampicillin. When the $OD_{600}$ reached 1.0, the cells were collected by centrifugation at 2000×$g$ and resuspended in 500 mL of M9 minimal medium with ampicillin, 0.4–1 mM $ZnSO_4$, $^{15}NH_4Cl$ with or without $^{13}C_6$-glucose for preparations of $^{13}C/^{15}N$-labeled or $^{15}N$-labeled samples, respectively. After shaking at 18 °C for an hour, isopropyl β-D-thiogalactoside (IPTG) was added to a final concentration of 0.4 mM to induce protein expression. After being induced for 18–20 h, the cells were centrifuged at 7000×$g$, resuspended in an appropriate buffer, and frozen at −80 °C. For WRKY33 samples, a 30 mM Tricine-NaOH buffer (pH 7.5) with 1 M NaCl was used, and 20 µM $ZnSO_4$ was added into the buffer for stabilization of the zinc-finger. For SIB1 constructs, a 20 mM Tris-HCl buffer (pH 8.5) with 1 M NaCl was used. Protein purifications were performed via Ni–NTA affinity chromatography followed by gel filtration (Superdex-75, GE Healthcare) chromatography.

## NMR spectroscopy
Protein samples were prepared in a buffer containing 30 mM MES (pH 6.0) and 50 mM NaCl. For WRKY33 constructs, 20 µM $ZnSO_4$ was added to the buffer to help stabilize the zinc-finger. $D_2O$ was added to the NMR sample for field lock, and sodium 2,2-dimethyl-2-silapentane-5-sulfonate was used as the internal chemical shift reference.

All NMR experiments were performed at 25 °C using Bruker Avance 500, 600, and 800 MHz spectrometers equipped with four RF channels and triple-resonance cryo-probes with pulsed field gradients. For chemical shift assignments of WRKY33_C and free SIB1, the two-dimensional (2D) $^{15}N$-edited HSQC and conventional 3D HNCA, HNCACB, CBCA(CO)NH, HNCO, HBHA(CO)NH, (H)CC(CO)NH, (H)CCH-TOCSY and H(C)CH-COSY experiments were performed. For chemical shift assignments of $^{13}C/^{15}N$-labeled SIB1 in complex with WRKY33_C, the 2D HSQC spectrum and 3D HNCA, HN(CO)CA, HNCACB, CBCA(CO)NH, HNCO, HN(CA)CO, HBHA(CO)NH, and (H)CC(CO)NH experiments were performed. All NMR spectra were processed using NMRPipe[28] and analyzed using NMRView[29].

The 2D HSQC experiments monitoring the interaction between different SIB1 peptides and $^{15}N$-labeled WRKY33_C were conducted at 25 °C using a buffer containing 30 mM MES (pH 6.0), 50 mM NaCl, 20 µM $ZnSO_4$, and $D_2O$ 10%. The spectrum of the $^{15}N$-labeled WRKY33_C alone at a concentration of 0.1 mM was recorded as the reference spectrum. Different SIB1 peptides were added to the $^{15}N$-WRKY33_C sample at a 2:1 molar ratio, and their HSQC spectra were recorded. For the $SIB1^{40-60}$ and $SIB1^{55-69}$ peptides, additional samples with peptide:WRKY33_C molar ratio of 5:1 were also prepared, and the HSQC spectra were acquired. CSPs of $^{15}N$-WRKY33_C upon binding to different SIB1 peptides were calculated using the empirical equation $\Delta\delta = \sqrt{\Delta\delta_H^2 + (\Delta\delta_N/6)^2}$, where $\Delta\delta_H$ and $\Delta\delta_N$ are the chemical shift changes in the $^1H$ and $^{15}N$ dimensions, respectively.

## Spin labeling and PRE experiments
The WRKY33_C D357C, R366C, and K376C mutants for spin labeling were expressed and purified similarly to the wild-type protein. The unlabeled WRKY33_C mutants were individually mixed with a fourfold molar excess of $Mn^{2+}$-chelated [N-(2-Maleimidoethyl)ethylenediamine-$N,N,N',N'$-tetraacetic acid, monoamide (Cat Number M138480, Toronto Research Chemicals, Inc.) and incubated for 4 h at room temperature. The spin-labeled sample was subsequently purified using a cation exchange column and buffer-exchanged into the NMR buffer (30 mM MES, 50 mM NaCl, pH 6.0).

For measurements of the intermolecular PRE data, spin-labeled WRKY33_C mutant samples (0.4 mM) were mixed with $^{15}N$-labeled SIB1 (0.6 mM). Because the HSQC spectra of $^{15}N$-labeled SIB1 in complex with wild-type WRKY33_C or with $Ca^{2+}$-chelated WRKY33_C mutants are essentially similar, the diamagnetic control experiment was recorded using the wild-type WRKY33_C following a similar strategy previously reported[30].

The PRE experiments were performed on a Bruker Avance 600 MHz spectrometer equipped with a cryogenic TCI probe. The transverse relaxation rates were measured for the paramagnetic and diamagnetic samples, and the $\Gamma_2$ values were calculated as the difference between the transverse relaxation rates of the diamagnetic and the paramagnetic samples[31].

## EMSA experiments
The oligonucleotides 5'- AAAGTTGACCAA-3' and 5'- TTGGTCAACTTT-3' were annealed to form the DNA duplex. The binding reactions (20 µl) were performed in the NMR buffer with 100 mM NaCl, using 1.0 ng double-stranded DNA with different concentrations of WRKY33_C, in the presence or absence of SIB1 (the molar ratio of SIB1: WRKY33_C was kept at 2:1 for each reaction). The binding reaction mixture was incubated at room temperature for 20 min, and the complex was separated from the free duplex by gel electrophoresis in 0.5 × TBE buffer (50 mM Tris base, 50 mM boric acid, 1 mM EDTA) at 20 mA for 50 min. The gel was stained with Gelred and the images were captured using a ChemiDocTM MP Imaging System (Bio-Rad), and the band intensities were read out for semi-quantifications.

## Chemical cross-linking experiments
Chemical cross-linking between WRKY33_C and SIB1 were carried out using the crosslinker EGS (Thermo Scientific) following the manufacturer's instructions. Prior to cross-linking, WRKY33_C and SIB1 were exchanged into a buffer containing 20 mM HEPES (pH 7.0) and 50 mM NaCl. The optimized WRKY33_C and SIB1 concentrations were 90 µM, and a 30-fold molar excess of the crosslinker was added to the sample. The reaction mixture was incubated at room temperature, and samples for Tricine-SDS-PAGE analysis were taken every 10 min. Finally, the reaction was quenched with a solution containing Tris at a final concentration of 20 mM.

## Size-exclusion chromatography
Size-exclusion chromatography (Superdex-75, GE Healthcare) was performed to analyze the complex formation between WRKY33_C and SIB1. A thioredoxin (Trx)-fused SIB1 construct (Trx-SIB1) was used to incubate with WRKY33_C prior to loading onto the column. The increased molecular weight of Trx (24 kDa) could help determine the binding stoichiometry with better accuracy.

## Structure modeling
All-atom molecular dynamics (MD) simulations were performed with the AMBER 16 package[32]. The starting conformation of WRKY33_C was generated by SWISS-MODEL[33] using the crystal structure of WRKY1 C-terminal DBD (PDB: 2AYD)[9] as the template, and the zinc metal center was patched as a Zn-CCHH type using Xplor-NIH[34]. For SIB1, only the S46–S80 segment was used and its initial conformation was also generated using AMBER 16. Inter-molecular distance restraints were generated based on the experimentally observed PRE values using the Solomon-Bloembergen equations[35]. A total of 17 PRE-derived restraints between the D357C /K376C sites in WRKY33_C and the corresponding residues in SIB1 exhibiting large PRE values, together with one mutagenesis-derived restraint between I358/I360 in WRKY33_C and V51 in SIB1, were added during the MD simulation. An energy penalty potential with a narrow flat region (1 Å) was used to account for the PRE restraints, and the force constants were set as 2.0 kcal/mol·Å. For the simulation, the AMBER ff14SB[36] force field was used for the protein, and the Zinc AMBER force field (ZAFF) was used for the zinc metal center[37]. The initial complex structure was solvated in a cubic TIP3P water box with a 10 Å padding for all directions. Four independent 500 ns MD simulation trajectories were performed at 298 K with a time step of 2 fs. The corresponding distances of all the snapshots from the simulation trajectories were calculated. Twenty conformers showing the best correlations with the experimental data were selected to represent the SIB1-WRKY33_C complex.

To build the SIB1-WRKY33_C-DNA ternary complex structure model, a representative conformer of the calculated SIB1-WRKY33_C

binary complex was aligned with the AtWRKY1-DNA complex (PDB: 6J4E)[13] to generate the initial structure. A randomization of this structure was performed in which only the N-terminus of SIB1 (residues prior to V51) was allowed to move freely. Multiple conformers with randomized SIB1 N-terminal conformations were selected, and the full N-terminal segment of SIB1 (residues T11-I45) was added to these structures. Subsequent structure calculation was performed using Xplor-NIH[34]. During the calculation, the backbone atoms of both WRKY33_C and the DNA were fixed, the V51-S80 segment of SIB1 was treated as a rigid body while the remaining regions were allow to move freely. To introduce structural restraints between the SIB1 K25-K32 segment and the DNA molecule, the collapse term that defines a radius of gyration restraint was added. A total of 120 structures were calculated and analyzed. The 20 lowest-energy conformers were selected as representative structural models.

## Statistics and reproducibility

The NMR titration experiments were repeated at least two times using different batches of protein samples. The EMSA and cross-linking experiments were repeated at least two times. The results were reproducible.

## Data availability

The chemical shift assignments of WRKY33_C and SIB1 in their free and complexed states have been deposited in the BioMagResBank (http://www.bmrb.wisc.edu/) under the accession numbers **50579**, **50580**, **50581**, **50582**, and **50583**. The structure of the SIB1-WRKY33_C complex has been deposited in the RCSB Protein Data Bank (https://www.rcsb.org/) under the accession number **8K31**. Other source data are provided in Supplementary Data 1.

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

## Acknowledgements

All NMR experiments were performed at the Beijing NMR Center, the NMR facility of the National Center for Protein Sciences at Peking University, and the National Center for Magnetic Resonance in Wuhan. We thank Prof. Bin Xia from Peking University for his kind discussions and suggestions. This work was supported by Grant 21991083 from the National Natural Science Foundation of China to Y.H., and Grant 2016YFA0501201 from the National Key R&D Program of China to C.J.

## Author contributions

C.J. and Y.H. conceived the project, designed experiments, and analyzed the data. X.D., L.Y. and Q.Z. conducted the NMR and biochemical experiments and analyzed the data. J.Y. contributed to the protein sample preparations. Z.G. contributed to the MD simulations. X.N. and H.L. contributed to NMR data collection and processing. X.Z. and M.L. contributed to data analysis and discussions. C.J. and Y.H. provided overall project supervision and critical suggestions. Y.H. and X.D. wrote the manuscript with contributions from all authors.

## Competing interests

The authors declare no competing interests.
