## [Peer review file · Communications Biology]

Reviewers' comments:

Reviewer #1 (Remarks to the Author):

The manuscript (COMMSBIO-23-3417) "Structural basis for the regulation of plant transcription factor WRKY33 by the intrinsically disordered VQ protein SIB1" employed NMR methods to investigate the interaction between Arabidopsis SIB1 and WRKY33 cDBD. This research has been planned and implemented very well and certain results presented here are useful.

WRKY33 plays an important role in plant stress resistance during growth and development, and the *in vivo* interaction between WRKY33 and SIB1 has been demonstrated for over a decade. This study further demonstrated their interaction relationship and possible spatial structures and conformations *in vitro*. I have the following questions:

1. The VQ motif of SIB1 is necessary for its interaction with WRKY33, and this study further proves its point. However, there are also many VQ protein VQ motifs that are not key sites for its interaction with other WRKYs. Teachers can explain the possible reasons from the perspective of this study
2. In this study, SIB1 is classified as the intrinsically ordered VQ protein (IDPs), which I personally recommend removing them as it is rarely mentioned in the text or in normal VQ protein function studies in plants.
3. The author discovered that the cluster II has essential contributions to interacting with the DNA molecule and stabilizing the SIB1-WRKY33_ C-DNA terminal complex. If it is possible to verify the function of cluster II *in vivo*, whether it affects the transcription regulation of WRKY33?
4. Line "It remains to be investigated whether SIB1 interact with WRKY75 in a similar way as WRKY33 and whether other VQ-WRKY complexes adopt different binding modes". Can the author make simple predictions through a certain model?
5. Line427, there is an extra comma, please delete it

Reviewer #2 (Remarks to the Author):

The authors use high resolution NMR spectroscopy and other biochemical methods (EMSA, SEC) to characterize the interaction between WRKY plant transcription factors and the protein SIB1. SIB1 is an IDP that binds to WRKY and regulates its DNA binding activity. The authors used NMR PRE experiments and MD simulations to generate a model for the WRKY-SIB1 complex. With these results they propose a mechanism by which a cluster of Lysine residues at the N-terminus of SIB stabilizes the binding of DNA by electrostatic interactions.

The paper is interesting as it provides new insights about the regulation of WRKY transcription factors and the molecular details that direct DNA binding. The experiments are of high quality but I

think that a few supporting experiments/controls are necessary before the work is suitable for publication.

- 1) One is about the effect of the EDTA tag on the structure of the WRYK. The authors use the untagged protein as diamagnetic control. They should demonstrate that the paramagnetic tag does not induce perturbations on protein structure and dynamics and use a diamagnetic metal (maybe Ca²⁺) in the tagged WRYK as a control.
- 2) It would be interesting to determine the K_ds of the different complexes, WRKY:SIB, WRK:DNA and WRK-SIB:DNA. Maybe the authors can already have an idea or make rough estimations based on the band intensity of the EMSA experiments? At least for the DNA interactions?
- 3) The model presented in Figure 7b is based on a previous structure of Arabidopsis WRYK DBD-DNA complex and as such and the role of SIB1 in that high resolution structural representation is not fully corroborated. Can the authors come up with a few experiments that would confirm this? Maybe titrating an oligonucleotide into the WRYK-SIB1 complex and measuring chemical shift perturbations or PRE effects that may confirm some of the features? The EMSA experiments with the Lysine mutants only shows that the cluster is necessary for the binding but does not provide structural or biophysical information of the interaction. Alternatively the authors could move the model to SI and finish the result section by just stressing the role of the cluster.

Reviewer #3 (Remarks to the Author):

With a combination of assays such as NMR, mutation, modeling, etc., Dong et al. present the interaction mechanism between WRKY33 DBD and its regulatory protein SIB1. The manuscript was well-written and plausible. Here are some suggestions.

1. Fig.1C, for cross-linking, a control with an irrelevant protein such as BSA is recommended.
2. Fig. 1D, a commercial protein standards with four to five components is recommended.
3. It would be better to provide some other experiment to show that Trx_SIB1 interact with WRKY33_C, such as BLI, pull-down, etc.

Response to reviewers' comments:

Reviewer #1 (Remarks to the Author):

The manuscript (COMMSBIO-23-3417) “Structural basis for the regulation of plant transcription factor WRKY33 by the intrinsically disordered VQ protein SIB1” employed NMR methods to investigate the interaction between Arabidopsis SIB1 and WRKY33 cDBD. This research has been planned and implemented very well and certain results presented here are useful.

WRKY33 plays an important role in plant stress resistance during growth and development, and the in vivo interaction between WRKY33 and SIB1 has been demonstrated for over a decade. This study further demonstrated their interaction relationship and possible spatial structures and conformations in vitro. I have the following questions:

1. The VQ motif of SIB1 is necessary for its interaction with WRKY33, and this study further proves its point. However, there are also many VQ protein VQ motifs that are not key sites for its interaction with other WRKYs. Teachers can explain the possible reasons from the perspective of this study.

Response: We thank the reviewer for the comment and suggestion. Our current results of the SIB1-WRKY33 system indicate that while the VQ motif is indispensable, it is not the only binding site necessary for interaction. The segment preceding this conserved motif was also found to be important for the complex formation. Our results are consistent with previous publications, e.g. the amino acids preceding the VQ motif are involved in interaction between the VQ protein and WRKY, determining affinity and specificity of WRKY-VQ binding (Zhou, Y. *et al.* 2016, *Sci Rep*, 6, 34663). In this sense, for other VQ proteins that interact with other WRKYs, sequences other than the VQ motif may play an important role in driving complex formation. We agree with the reviewer that these are interesting questions and the detailed mechanism remains to be investigated. However, since we do not have experimental data related to other VQ-WRKY pairs, we feel that it might be better that we keep focus on the SIB1-WRKY33 system in the current manuscript.

In the revised manuscript, we have added a sentence of discussion and cited the above reference (see page 23, line 513-515).

2. In this study, SIB1 is classified as the intrinsically ordered VQ protein (IDPs), which I personally recommend removing them as it is rarely mentioned in the text or in normal VQ protein function studies in plants.

Response: We thank the reviewer for the suggestion. We have modified the title of the revised manuscript to remove the phrase ‘intrinsically disordered’ following the reviewer’s suggestion, while keeping the phrase in the main text of the manuscript. Even though the phrase is seldom mentioned in previous functional studies, we believe that it could be helpful important to point out that the VQ proteins fall into the category of IDPs, because IDPs can behave rather differently from well-structured proteins and strongly contribute to the dynamics (or flexibility) of the VQ-WRKY

complexes. To better present this, we made a small modification in the Introduction section of the revised manuscript, in both line 62 and line 70-71 as follows:

Line 62: “A growing body of evidence highlights a class of VQ proteins, to act as transcriptional regulators of WRKY TFs (17-20).”

Line 70-71: “Up to date, about 34 VQ proteins have been identified in *Arabidopsis* and they belong to the intrinsically disordered proteins (IDPs) based on sequence analysis.”

3. The author discovered that the cluster II has essential contributions to interacting with the DNA molecule and stabilizing the SIB1-WRKY33_ C-DNA terminal complex. If it is possible to verify the function of cluster II in vivo, whether it affect the transcription regulation of WRKY33?

Response: We appreciate the reviewer’s suggestion on *in vivo* assay, and we also believe that the novelty of our study could be improved if it can be performed. However, we are afraid that that it is technically not feasible in our laboratory.

4. Line “It remains to be investigated whether SIB1 interact with WRKY75 in a similar way as WRKY33 and whether other VQ-WRKY complexes adopt different binding modes”. Can the author make simple predictions through a certain model?

Response: We thank the reviewer for the interesting question. We think it possible that the VQ motif may have similar packing as seen in the SIB1-WRKY33 complex model, while the flanking regions may have similar or different interaction modes. However, this is the first structural study of VQ-WRKY interactions and much remains unknown at the current stage. Without convincing experimental data, we don’t wish to make more predictions that could be misleading.

5. Line427, there is an extra comma, please delete it

Response: We thank the reviewer very much for pointing out our mistake. We have corrected it in the revised manuscript.

Reviewer #2 (Remarks to the Author):

The authors use high resolution NMR spectroscopy and other biochemical methods (EMSA, SEC) to characterize the interaction between WRYK plant transcription factors and the protein SIB1. SIB1 is an IDP that binds to WRYK and regulates its DNA binding activity. The authors used NMR PRE experiments and MD simulations to generate a model for the WRYK-SIB1 complex. With these results they propose a mechanism by which a cluster of Lysine residues at the N-terminus of SIB1 stabilizes the binding of DNA by electrostatic interactions.

The paper is interesting as it provides new insights about the regulation of WRYK transcription factors and the molecular details that direct DNA binding. The experiments are of high quality but I think that a few supporting experiments/controls are necessary before the work is suitable for publication.

1) One is about the effect of the EDTA tag on the structure of the WRYK. The authors use the untagged protein as a diamagnetic control. They should demonstrate that the paramagnetic tag does not induce perturbations on protein structure and dynamics and use a diamagnetic metal (maybe Ca^{2+}) in the tagged WRYK as a control.

Response: We thank the reviewer very much for the suggestion. Following the reviewer's suggestion, we prepared some more ^{15}N -labeled SIB1 and EDTA-tagged WRKY33_C samples (the D357C and R366C sites were tested) during the revision and recorded the Ca^{2+} -chelated spectra as control experiments. The HSQC spectra of SIB1 complexed with the diamagnetic Ca^{2+} -chelated WRKY33 sample are essentially similar to the spectrum obtained in the complex with wild-type WRKY33 (Response Fig. 1a-c). Moreover, the normalized amide signal intensity profiles are also essentially similar between the tagged and the wild-type samples (Response Fig. 1d-e). These results indicate that EDTA-tagging does not have an obvious influence on the native interaction and dynamics between WRKY33 and SIB1. Therefore, using either the Ca^{2+} -chelated spectra or the wild-type protein spectra as a control would not change the resulting PRE profiles or the distance restraints used for building the complex model.

In the revised manuscript, we modified the 'Spin labeling and PRE experiments' part of the Methods section to include a short description of this issue, together with citation of a previous literature that also used untagged proteins as control experiments (page 6, line 141-144 of the revised manuscript).

Response Figure 1. (a-c) The ¹H-¹⁵N HSQC spectra of ¹⁵N-labeled SIB1 with wild type WRKY33_C, Ca²⁺-EDTA tagged D356C WRKY33_C and Ca²⁺-EDTA tagged R366C WRKY33_C. (d-f) Normalized backbone amide signal intensity profiles of the ¹⁵N-labeled SIB1 samples in complex with wild-type WRKY33_C or with Ca²⁺-EDTA tagged WRKY33_C mutants.

2) It would be interesting to determine the *K*_ds of the different complexes, WRKY:SIB, WRK:DNA and WRK-SIB:DNA. Maybe the authors can already have an idea or make rough estimations based on the band intensity of the EMSA experiments? At least for the DNA interactions?

Response: We appreciate the reviewer's suggestion and we agree that *K*_d determination would be an interesting point. Based on the EMSA experiments, we could estimate the WRKY33-DNA *K*_d to be in the sub-micromolar range (~ 150 μM) based on band intensities (Response Fig. 2b). However,

when we tried to use the EMSA experiments to estimate the K_d between DNA and the WRKY33-SIB1 complex, we were met with some difficulties. As shown in Response Fig. 2, we can observe that the fractions of bound state of DNA reach at 1:1 molar ratio close to 95% for both WRKY33/SIB1 and WRKY33/SIB1^{mini}, compared to ~ 80% for WRKY33 alone, suggesting enhanced DNA binding in the presence of SIB1. However, the curves obtained for the WRKY33-SIB1 complexes fitted poorly to a two-state exchange model and therefore the derived K_d values could be problematic. We think that the problem may be due to the partial dissociation between WRKY33 and SIB1, as indicated by the presence of the band corresponding to WRKY33-DNA binary complex in Response Fig. 2 and in Figure 1B of the main text. This band is also often visible in other trials that we have conducted, and its amount also varies depending on the sample batch, the voltage used during the electrophoresis, etc. We found it very difficult to eliminate or control the appearance/amount of this dissociation product. This may be the cause for the complexity of the apparent binding curve and explain why the curve could not be well-fitted.

In addition, we have also tried to use ITC experiments for K_d measurements. However, severe precipitation was observed after addition of DNA into the WRKY33-SIB1 complex sample at the required protein concentrations. The precipitation was identified to contain all three components, and we suppose that the ternary complex may not have enough solubility to allow for ITC experiments.

Due to these complexities for the ternary system, we are not able to estimate a K_d between the DNA and the WRKY33-SIB1 complex with accuracy. Therefore, we think it may be better to only draw qualitative conclusions regarding the interactions in our current manuscript.

Response Figure 2. The EMSA assays detecting the interaction between WRKY33/SIB1 and DNA. (a) The gel-image of EMSA assay. (b-d) The curves show the fitting of the apparent K_d values based on band brightness.

3) The model presented in Figure 7b is based on a previous structure of Arabidopsis WRYK DBD-DNA complex and as such and the role of SIB1 in that high resolution structural representation is not fully corroborated. Can the authors come up with a few experiments that would confirm this? Maybe titrating an oligonucleotide into the WRYK-SIB1 complex and measuring chemical shift perturbations or PRE effects that may confirm some of the features? The EMSA experiments with the Lysine mutants only shows that the cluster is necessary for the binding but does not provide structural or biophysical information of the interaction. Alternatively the authors could move the model to SI and finish the result section by just stressing the role of the cluster.

Response: We thank the reviewer very much for the comments and suggestions. We are very interested in examining the ternary complex packing by experimental means. Actually, we have made a lot of efforts in trying to get an NMR sample of the WRKY33-SIB1-DNA complex. However, this ternary complex undergoes significant precipitation at higher protein concentrations (typically in the hundreds of micromolar to millimolar range for NMR experiments), making NMR measurements impossible. This phenomenon also hinders ITC measurement as describe above. We have tried optimizing the buffer conditions, etc, but have not succeeded in solving this problem.

Hence, we are not able to provide further experimental support for the ternary structure model at the current stage. Following the reviewer's suggestion on rearranging the figures, we have moved the figure panel of the ternary structure model into the SI file (Supplemental Fig. S13), and also moved the text related to this model into the Discussion section (see page 24, line 530-541 of the revised manuscript).

Reviewer #3 (Remarks to the Author):

With a combination of assays such as NMR, mutation, modeling, etc., Dong et al. present the interaction mechanism between WRKY33 DBD and its regulatory protein SIB1. The manuscript was well-written and plausible. Here are some suggestions.

1. Fig. 1C, for cross-linking, a control with an irrelevant protein such as BSA is recommended.

Response: We thank the reviewer for the suggestion. In Response Figure 3 below, we show the cross-linking experiments between BSA and WRKY33_C, and between BSA and SIB1, and we do not observe strong bands for non-specific cross-linking products. Moreover, we used the cross-linking between WRKY33_C and SIB1 primarily for verifying the binding stoichiometry. The specific binding between the two proteins can be established more directly from the NMR results.

Response Figure 3. The image of SDS-PAGE showing the EGS cross-linking reaction between WRKY33_C/SIB1¹¹⁻¹⁰⁰ and BSA.

2. Fig. 1D, a commercial protein standards with four to five components is recommended.

Response: We thank the reviewer for the suggestion and we agree that using commercial protein standards with more components would be better. The SEC assay presented in our manuscript was calibrated using a commercially bought standard globular proteins. However, these standard proteins were bought separately rather than as a mixture. As an NMR laboratory, we commonly work with small proteins ($M_w < 30$ kDa) and using these low molecular-weight protein standards could generally cover our needs. In the case of the Trx-SIB1/WRKY33 complex, its size falls in-between the sizes of the bovine carbonic anhydrase (31 kDa) and rabbit actin protein (43 kDa). We believe that this result meets the need of the experiment, while adding more standard proteins would not essentially affect the conclusion.

3. *It would be better to provide some other experiment to show that Trx_SIB1 interact with WRKY33_C, such as BLI, pull-down, etc.*

Response: We appreciate the reviewers suggestion. We added the HSQC spectra showing that the Trx_SIB1 interact with WRKY33_C in the same way as wt-SIB1 in the Supporting Information of the revised manuscript (Supplemental Fig. S2 of the revised manuscript).

Response to reviewers' comments:

Reviewer #1 (Remarks to the Author):

The manuscript (COMMSBIO-23-3417) “Structural basis for the regulation of plant transcription factor WRKY33 by the intrinsically disordered VQ protein SIB1” employed NMR methods to investigate the interaction between Arabidopsis SIB1 and WRKY33 cDBD. This research has been planned and implemented very well and certain results presented here are useful.

WRKY33 plays an important role in plant stress resistance during growth and development, and the in vivo interaction between WRKY33 and SIB1 has been demonstrated for over a decade. This study further demonstrated their interaction relationship and possible spatial structures and conformations in vitro. I have the following questions:

1. The VQ motif of SIB1 is necessary for its interaction with WRKY33, and this study further proves its point. However, there are also many VQ protein VQ motifs that are not key sites for its interaction with other WRKYs. Teachers can explain the possible reasons from the perspective of this study.

Response: We thank the reviewer for the comment and suggestion. Our current results of the SIB1-WRKY33 system indicate that while the VQ motif is indispensable, it is not the only binding site necessary for interaction. The segment preceding this conserved motif was also found to be important for the complex formation. Our results are consistent with previous publications, e.g. the amino acids preceding the VQ motif are involved in interaction between the VQ protein and WRKY, determining affinity and specificity of WRKY-VQ binding (Zhou, Y. *et al.* 2016, *Sci Rep*, 6, 34663). In this sense, for other VQ proteins that interact with other WRKYs, sequences other than the VQ motif may play an important role in driving complex formation. We agree with the reviewer that these are interesting questions and the detailed mechanism remains to be investigated. However, since we do not have experimental data related to other VQ-WRKY pairs, we feel that it might be better that we keep focus on the SIB1-WRKY33 system in the current manuscript.

In the revised manuscript, we have added a sentence of discussion and cited the above reference (see page 24, line 519-521) as follows:

“Consistent with our results, experimental evidence supporting the importance of the segment preceding the VQ motif has also been reported for soybean VQ proteins (35).”

2. In this study, SIB1 is classified as the intrinsically ordered VQ protein (IDPs), which I personally recommend removing them as it is rarely mentioned in the text or in normal VQ protein function studies in plants.

Response: We thank the reviewer for the suggestion. We have modified the title of the revised manuscript to remove the phrase ‘intrinsically disordered’ following the reviewer’s suggestion, while keeping the phrase in the main text of the manuscript. Even though the phrase is seldom mentioned in previous functional studies, we believe that it could be helpful important to point out

that the VQ proteins fall into the category of IDPs, because IDPs can behave rather differently from well-structured proteins and strongly contribute to the dynamics (or flexibility) of the VQ-WRKY complexes. To better present this, we made a small modification in the Introduction section of the revised manuscript, in both line 62 and line 70-72 as follows:

Line 62: “A growing body of evidence highlights a class of VQ proteins, to act as transcriptional regulators of WRKY TFs (17-20).”

Line 70-72: “Up to date, about 34 VQ proteins have been identified in *Arabidopsis thaliana*. They all display sequence features characteristic of intrinsically disordered proteins (IDPs).”

3. *The author discovered that the cluster II has essential contributions to interacting with the DNA molecule and stabilizing the SIB1-WRKY33_C-DNA terminal complex. If it is possible to verify the function of cluster II in vivo, whether it affect the transcription regulation of WRKY33?*

Response: We appreciate the reviewer’s suggestion on *in vivo* assay, and we also believe that the novelty of our study could be improved if it can be accomplished. While we regret to say that the assay is technically not feasible in our laboratory, we added a brief proposal in the Discussion section of the revised manuscript to address this issue as follows:

“While our current study offers a structural-based hypothesis for the function of the cluster II lysines in promoting a ternary complex formation, we anticipate that future *in vivo* studies, e.g. *in vivo* transcriptional activity assays, or functional examinations of *Arabidopsis* resistance to *B. cinerea* infection using transgenic plants harboring SIB1/2 mutants as described by Lai *et al* (21), to be essential in providing further insights into the physiological role of the lysine cluster.” (Page 25, line 558-562)

4. *Line “It remains to be investigated whether SIB1 interact with WRKY75 in a similar way as WRKY33 and whether other VQ-WRKY complexes adopt different binding modes”. Can the author make simple predictions through a certain model?*

Response: We thank the reviewer for the interesting suggestion. Following the reviewer’s suggestion, we built a structural model of WRKY75 DNA binding domain and compared it with the SIB1-WRKY33_C complex (Response Figure 1). Based on the model, we think it possible that SIB1/2 may interact with WRKY75 at the same site and via a similar interaction mode as WRKY33_C. To discuss this possibility, we added the Response Figure 1 as Supplemental Fig. S16 in the SI file of the revised manuscript, and also added a short paragraph in the Discussion section as follows:

“By comparing the structural model of the WRKY75 DNA-binding domain with the SIB1-WRKY33_C complex structure, we observe that WRKY75 exhibits a generally similar electrostatic distribution pattern as WRKY33_C. Additionally, WRKY75 contains two hydrophobic residues, Val63 and Ile65, at positions corresponding to I358 and I360 of WRKY33 (**Supplemental Fig. S16**).

These similar characteristics suggest that SIB1/2 may also interact with WRKY75 at the same site and probably via an analogous binding pattern. However, the exact binding mode as well as the mechanisms of how SIB1/2 contributes to the regulation of different signaling pathways remain to be investigated.” (Page 25-26, line 566-573)

Response Figure 1. (Figure S16 of the revised manuscript) Structural comparison between WRKY33_C and WRKY75.

5. Line427, there is an extra comma, please delete it

Response: We thank the reviewer very much for pointing out our mistake. We have corrected it in the revised manuscript.

Reviewer #2 (Remarks to the Author):

The authors use high resolution NMR spectroscopy and other biochemical methods (EMSA, SEC) to characterize the interaction between WRKY plant transcription factors and the protein SIB1. SIB1 is an IDP that binds to WRKY and regulates its DNA binding activity. The authors used NMR PRE experiments and MD simulations to generate a model for the WRKY-SIB1 complex. With these results they propose a mechanism by which a cluster of Lysine residues at the N-terminus of SIB1 stabilizes the binding of DNA by electrostatic interactions.

The paper is interesting as it provides new insights about the regulation of WRKY transcription factors and the molecular details that direct DNA binding. The experiments are of high quality but I think that a few supporting experiments/controls are necessary before the work is suitable for

publication.

1) One is about the effect of the EDTA tag on the structure of the WRKY. The authors use the untagged protein as diamagnetic control. They should demonstrate that the paramagnetic tag does not induce perturbations on protein structure and dynamics and use a diamagnetic metal (maybe Ca^{2+}) in the tagged WRKY as a control.

Response: We thank the reviewer very much for the suggestion. Following the reviewer's suggestion, we prepared some more ^{15}N -labeled SIB1 and EDTA-tagged WRKY33_C samples (the D357C and R366C sites were tested) during the revision and recorded the Ca^{2+} -chelated spectra as control experiments. The HSQC spectra of SIB1 complexed with the diamagnetic Ca^{2+} -chelated WRKY33 sample are essentially similar to the spectrum obtained in the complex with wild-type WRKY33. Moreover, the normalized amide signal intensity profiles are also essentially similar between the tagged and the wild-type samples (see Response Figure 2). These results indicate that EDTA-tagging does not have obvious influence the native interaction and dynamics between WRKY33 and SIB1. Therefore, using either the Ca^{2+} -chelated spectra or the wild-type protein spectra as a control would not change the resulting PRE profiles or the distance restraints used for building the complex model.

Response Figure 2. (Figure S8 of the revised manuscript) Spectral comparison of ^{15}N -labeled SIB1 in complex with wild-type WRKY33_C or Ca^{2+} -EDTA-tagged WRKY33_C mutants.

In the revised manuscript, we also added the above results as Supplementary Fig. S8 in the SI file and modified the ‘Spin labeling and PRE experiments’ part of the Methods section to include a short description of this issue, together with citation of a previous literature that also used untagged proteins as control experiments (page 6, line 141-144 of the revised manuscript).

2) *It would be interesting to determine the K_d s of the different complexes, WRKY:SIB, WRK:DNA and WRK-SIB:DNA. Maybe the authors can already have an idea or make rough estimations based on the band intensity of the EMSA experiments? At least for the DNA interactions?*

Response: We appreciate the reviewer’s suggestion and we agree that K_d determination would be an interesting point. Based on the EMSA experiments, we could estimate the WRKY33-DNA K_d to be in the sub-micromolar range (~ 150 nM) based on band intensities. However, when we tried to use the EMSA experiments to estimate the K_d between DNA and the WRKY33-SIB1 complex, we were met with some difficulties. As shown in Response Figure 3, we can observe that at 1:1 molar ratio, the fractions of bound state of DNA reach close to 95% for WRKY33/SIB1, compared to $\sim 80\%$ for WRKY33 alone, suggesting enhanced DNA binding in the presence of SIB1. However, the curves obtained for the WRKY33-SIB1 complexes fitted poorly to a two-state exchange model and therefore the derived K_d values could be problematic. We think that the problem may be due to the partial dissociation between WRKY33 and SIB1, as indicated by the presence of the band corresponding to WRKY33-DNA binary complex (see Response Figure 3 and also Figure 1B of the main text). This band is also often visible in other trials that we have conducted, and its amount also varies depending on the sample batch, the voltage used during the electrophoresis, etc. We found it very difficult to eliminate or control the appearance/amount of this dissociation product. This may be the cause for the complexity of the apparent binding curve and explain why the curve could not be well-fitted.

In addition, we have also tried to use ITC experiments for K_d measurements. However, severe precipitation was observed after addition of DNA into the WRKY33-SIB1 complex sample at the required protein concentrations. The precipitation was identified to contain all three components, and we suppose that the ternary complex may not have enough solubility to allow for ITC experiments. Due to these complexities for the ternary system, we are not able to estimate a K_d between the DNA and the WRKY33-SIB1 complex with accuracy. Therefore, we think it may be better to only draw qualitative conclusions regarding the interactions in our current manuscript.

To better address these issues, we have included the above figure as Supplemental Fig. S15 in the revised SI file, and added a paragraph in the Discussion section as follows:

“Based on the EMSA results, we could estimate that the binding between WRKY33_C and the W-box DNA has a dissociation constant K_d in the sub-micromolar range (~ 150 nM). However, in the case of the binding between the SIB1-WRKY33_C binary complex and DNA, the binding curves deviate from a two-state exchange model probably due to the partial dissociation of the SIB1-WRKY33_C complex (**Fig. 1B** and **Supplemental Fig. S15**). Furthermore, we noted that the SIB1-

WRKY33_C-DNA ternary complex tends to precipitate at higher concentrations (e.g., sub-millimolar to millimolar range), thereby preventing the use of isothermal titration calorimetry or solution NMR techniques for further characterization of the binding. Therefore, we were not able to provide an accurate estimation of the binding affinity under the current experimental conditions.” (Page 25, line 549-558)

Response Figure 3. (Figure S15 of the revised manuscript) EMSA assays for investigating the interactions between WRKY33_C-DNA and WRKY33_C/SIB1-DNA.

3) The model presented in Figure 7b is based on a previous structure of Arabidopsis WRYK DBD-DNA complex and as such and the role of SIB1 in that high resolution structural representation is not fully corroborated. Can the authors come up with a few experiments that would confirm this? Maybe titrating an oligonucleotide into the WRYK-SIB1 complex and measuring chemical shift perturbations or PRE effects that may confirm some of the features? The EMSA experiments with the Lysine mutants only shows that the cluster is necessary for the binding but does not provide structural or biophysical information of the interaction. Alternatively the authors could move the model to SI and finish the result section by just stressing the role of the cluster.

Response: We thank the reviewer very much for the comments and suggestions. We are very interested in examining the ternary complex packing by experimental means. Actually, we have made a lot of efforts in trying to get an NMR sample of the WRKY33-SIB1-DNA complex.

However, this ternary complex undergoes significant precipitation at higher protein concentrations (typically in the hundreds of micromolar to millimolar range for NMR experiments), making NMR measurements impossible. This phenomenon also hinders ITC measurement as describe above. We have tried optimizing the buffer conditions, etc, but have not succeeded in solving this problem. Hence, we are not able to provide further experimental support for the ternary structure model at the current stage. Following the reviewer's suggestion on rearranging the figures, we have moved the figure panel of the ternary structure model into the SI file (Supplemental Fig. S13), and also moved the text related to this model into the Discussion section (see page 24-25, line 534-548 of the revised manuscript).

Reviewer #3 (Remarks to the Author):

With a combination of assays such as NMR, mutation, modeling, etc., Dong et al. present the interaction mechanism between WRKY33 DBD and its regulatory protein SIB1. The manuscript was well-written and plausible. Here are some suggestions.

1. Fig.1C, for cross-linking, a control with an irrelevant protein such as BSA is recommended.

Response: We thank the reviewer for the suggestion. We performed the cross-linking experiments between BSA and WRKY33_C as the reviewer suggested and we do not observe strong bands for non-specific cross-linking products (Response Figure 4). The results are also added as Supplemental Fig. S2A in the SI file of the revised manuscript. Moreover, we used the cross-linking between WRKY33_C and SIB1 primarily for verifying the binding stoichiometry. The specific binding between the two proteins can be established more directly from the NMR results.

Response Figure 4. (Figure S2A of the revised manuscript) Control experiment showing the EGS cross-linking results between WRKY33_C/SIB1¹¹⁻¹⁰⁰ and BSA.

2. *Fig. 1D, a commercial protein standards with four to five components is recommended.*

Response: We thank the reviewer for the suggestion and we agree that using commercial protein standards with more components would be better. The SEC assay presented in our manuscript was calibrated using a commercially bought standard globular proteins. However, these standard proteins were bought separately rather than as a mixture. As an NMR laboratory, we commonly work with small proteins (Mw < 30 kDa) and using these low molecular-weight protein standards could generally cover our needs. In the case of the Trx-SIB1/WRKY33 complex, its size falls in-between the sizes of the bovine carbonic anhydrase (31 kDa) and rabbit actin protein (43 kDa). We believe that this result meets the need of the experiment, while adding more standard proteins would not essentially affect the conclusion.

3. *It would be better to provide some other experiment to show that Trx_SIB1 interact with WRKY33_C, such as BLI, pull-down, etc.*

Response: We appreciate the reviewer's suggestion. We added the HSQC spectra showing that the Trx_SIB1 interact with WRKY33_C in the same way as wt-SIB1 in the Supporting Information of the revised manuscript (Supplemental Fig. S2B of the revised manuscript).

REVIEWERS' COMMENTS:

Reviewer #1 (Remarks to the Author):

The author has basically answered my previous question and is now suitable for publication. If conditions permit, it is recommended that this type of protein interaction experiment can be combined with plant in vivo experiments.

Reviewer #2 (Remarks to the Author):

In the revised version of the manuscript the authors provided reasonable answers to my concerns. Either by performing new experiments/analysis or by discussing the experimental difficulties associated with acquiring new data (direct binding and structural information of the ternary complex).

I believe the paper is suitable for publication in Communications Biology in its present form.

Reviewer #3 (Remarks to the Author):

The reviewer has no other question.

RESPONSE TO REVIEWERS' COMMENTS:

Reviewer #1 (Remarks to the Author):

The author has basically answered my previous question and is now suitable for publication. If conditions permit, it is recommended that this type of protein interaction experiment can be combined with plant in vivo experiments.

Response: We thank the reviewer very much for the comments and suggestions.

Reviewer #2 (Remarks to the Author):

In the revised version of the manuscript the authors provided reasonable answers to my concerns. Either by performing new experiments/analysis or by discussing the experimental difficulties associated with acquiring new data (direct binding and structural information of the ternary complex).

I believe the paper is suitable for publication in Communications Biology in its present form.

Response: We thank the reviewer very much for the comments and suggestions.

Reviewer #3 (Remarks to the Author):

The reviewer has no other question.

Response: We thank the reviewer very much for the comments and suggestions.